# Extracting and Analyzing Pyrrolizidine Alkaloids in Medicinal Plants: A Review

**DOI:** 10.3390/toxins12050320

**Published:** 2020-05-13

**Authors:** Thomas Kopp, Mona Abdel-Tawab, Boris Mizaikoff

**Affiliations:** 1Department of Chemistry, Institute of Analytical and Bioanalytical Chemistry, Ulm University, 89081 Ulm, Germany; Kopp.Thomas1984@gmx.de; 2Central Laboratory of German Pharmacists, 65760 Eschborn, Germany; m.tawab@zentrallabor.com

**Keywords:** pyrrolizidine alkaloids, analytical techniques, extraction techniques, medicinal plants, *Symphytum*, *Tussilago*, *Senecio*

## Abstract

Pyrrolizidine alkaloids (PAs) are distributed in plant families of *Asteraceae*, *Boraginaceae,* and *Fabaceae* and serve in the chemical defense mechanism against herbivores. However, they became a matter of concern due to their toxicity associated with the high risk of intake within herbal preparations, e.g., phytopharmaceutical formulations, medicinal teas, or other plant-derived drug products. In 1992, the German Federal Ministry of Health established the first limits of PA content for fourteen medicinal plants. Because of the toxic effects of PAs, the Federal Institute of Risk Assessment (BfR) established more stringent limits in 2011, whereby a daily intake <0.007 µg/kg body weight was recommended and valid until 2018. A threefold higher limit was then advised by BfR. To address consumer safety, there is the need for more efficient extraction procedures along with robust, selective, and sensitive analytical methods to address these concerns. With the increased prevalence of, e.g., phytopharmaceutical formulations, this timely review comprehensively focuses on the most relevant extraction and analysis strategies for each of those fourteen plant genera. While a variety of extraction procedures has been reported, differences in PA content of up to 1110 ppm (0.11% (*w*/*w*)) were obtained dependent on the nature of the solvent and the applied extraction technique. It is evident that the efficient extraction of PAs requires further improvements or at least standardization of the extraction conditions. Comparing the various analytical techniques applied regarding selectivity and sensitivity, LC-MS methods appear most suited. This review shows that both standardized extraction and sensitive determination of PAs is required for achieving appropriate safety levels concerning public health in future.

## 1. Introduction

Pyrrolizidine alkaloids (PAs) are secondary plant constituents produced by a wide variety of plants (*Asteraceae*, *Boraginaceae,* and *Fabaceae*) [1,2]. More than 500 different PAs were identified in more than 6000 plant species to date [2]. They are believed to function against herbivores due to the proven inhibiting effect of some PAs on acetylcholinesterase activity [3,4,5,6]. Recently, PAs and their corresponding N-oxides (PANs) have been discussed controversially due to their various toxic effects, induced by metabolic activation, resulting in tumors and necrosis of tissue [1,7,8,9,10]. The first limits were set in 1992 by the German Federal Ministry of Health addressing pharmaceutical products containing herbal preparations of PA producing plants. In more detail, herbs of the genera *Symphytum, Borago, Brachyglottis, Cineraria, Alkanna, Tussilago, Lithospermum, Cynoglossum, Senecio, Eupatorium, Anchusa, Petasites, Heliotropium*, and *Erechthites* [11] were concerned. It was proposed that exposure may not exceed 1 µg/day during a time period of six weeks. In case of longer usage, this level was further reduced to 0.1 µg/day. PA contents which lead to an increased exposure (10 µg/day) were accepted for herbal infusions or decoctions of *Tussilago farfara*. However, the high risk for ingesting these toxic substances was determined to not only originate from PA-producing medicinal plants. Herbal preparations may be contaminated with weeds as a result of disregardful harvesting or insufficient removal during further processing. Due to the risk of intake and the high toxicity, the Federal Institute of Risk Assessment in Germany (BfR) recommended a daily intake of not more than 0.007 µg/kg body weight (i.e., 0.42 µg for a 60 kg human) [2] in 2011. With respect to the limit stated by German Health Authority in 1992 (1 µg/day), this correspond to a 50% lower threshold. New risk assessments of European Food Safety Agency (EFSA) and BfR were based on the NTP study on riddelliine rather than the one on lasiocarpine and derived a threefold higher reference point of 237 µg/kg bw/day. This effectively means that an exposure lower than 0.0237 µg/kg bw/day is considered a low health concern (margin of exposure larger than 10,000) [12,13]. Especially for the required detection of the low levels of PAs, more efficient extraction methods along with selective and sensitive analytical strategies are essential. 

This review is dedicated to summarizing the scientific progress within approximately 30 years of research in PA extraction and analysis of medicinal plants. For each of them, we summarize the most prevalent extraction techniques and analytical methods used for determining their concentration levels. Parameters such as increased extraction temperatures, fresh solvent, high pressure, and a large surface area are compared for promoting efficient extraction. Analytical methods are rated according the reported selectivity of the procedure and the analytical-figures-of-merit with special emphasis on the sensitivity (LoD and LoQ). Additionally, whenever reasonable, the potential advantages and disadvantages of the analytical method are outlined. 

## 2. Design and Methods

A systematic literature research was performed using the Mendeley online database (www.mendeley.com). Keywords for the search were “plant genus + pyrrolizidine” or “plant genus + medicinal use”; for example, “*Senecio* + pyrrolizidine” or “*Senecio* + medicinal use”. All published manuscripts regarding extraction or analysis of PAs were reviewed. Results were not filtered by year or language. Published literature was considered if extractions or analytical studies were performed on one of the thirteen medicinal plant genera of interest that have been defined by the German Federal Ministry of Health. Finally, more than 225 papers were considered for this review.

## 3. Considered Medicinal Plants

The German Federal Ministry of Health, addressed in 1992 the admission and registration of herbal preparations containing PAs [11]. Fourteen plants were considered to be of phytopharmaceutical relevance. These plants were reviewed for the extraction and analytical methods applied. Except the genus *Erechthites*, all other genera were represented in scientific literature. Examples of inherent PAs are shown in Figure 1, Figure 2 and Figure 3. Because of the large number of species that have been partly investigated and the differences in the PA composition depending on the growth conditions and time of investigations of the individual species, only the genera are cited.

## 4. Extraction, and Analytical Techniques

Various techniques for extraction (see Table 1) of pyrrolizidine alkaloids from medicinal plants (see Table 2) are discussed. While some of them are more commonly used than others, most of them are based on maceration or percolation, the two main extraction principles stated in the literature (see Table 3). During maceration, the sample is soaked continuously with solvent, whereas, during percolation, the solvent flows through the plant material. Due to the dependence of the extraction yield on parameters such as viscosity of the solvent, diffusion coefficient and sample penetration, the technical implementation and the physical conditions vary.

After successful extraction, sample solutions were most commonly analyzed using chromatographic separation techniques such as HPLC, TLC and GC coupled with various detection methods. Capillary electrophoresis (CE) with the separation based on migration of molecules in an electric field, droplet counter current chromatography (DCCC) serving as an automated liquid–liquid separation method, and miscellaneous electrokinetic chromatography (MEKC) techniques, which are based on a combination of electrophoretic and chromatographic principles, were used as well. Additionally, specific techniques without prior separation or purification such as ^1^H-NMR, ^13^C-NMR, or ELISA (Enzyme Linked Immunosorbent Assay) tests based on specific antibody interactions with the target molecule were used for quantification. An overview of the separation and detection techniques is given in Table 4.

## 5. Results

Thirteen of the fourteen genera relevant in 1992 are considered in the following sections. The genus *Erechtites* could ultimately not be considered because of a lack of published data.

### 5.1. Alkanna

The genus *Alkanna* includes about 50 subtypes. For example, *Alkanna tinctorial* is used topically for skin treatment due to its antibacterial effects [172]. The PAs found in the genus *Alkanna* are listed in Table 2. Extraction was done by grinding the plant parts in Ultraturax using 0.5 N hydrochloric acid as a solvent followed by soaking for 1 h [14]. Another procedure uses maceration in methanol [15] (see Table 3). Quantitative analysis was performed by GC-MS [14]. Identification or qualitative analysis was performed by ^13^C-NMR, ^1^H-NMR, infrared spectroscopy, or mass spectrometry [15] (see Table 4).

### 5.2. Anchusa

This genus includes about 50 species and is used in folk medicine for treating open wounds and cuts [173]. Inherent PAs (see Table 2) were extracted by stepwise maceration with methanol and CHCl_3_ [123,124] by grinding plant parts in 0.5 N HCl, as described by El Shazly [14] or using an acidified mixture of methanol, combining the extraction power of organic, aqueous and acidic solvents [125] (see Table 3). Analytical methods (see Table 4) were focused on isolation and identification of the different PAs/PANs predominantly by GC-MS [14,16,124]. Siciliano et al. illustrated the PA pattern in leaves, flowers and roots applying LC-MS. They were able to show that the leaves contain most PAs followed by flowers and roots [123]. These et al. applied the previously reported LC-MS/MS screening method [125].

### 5.3. Borago

Plants of the genus *Borago* are used in herbal medicinal products due its antihypertensive effect [174], amoebicidal activity to treat gastrointestinal diseases [175], and antioxidant and antimicrobial effects [176,177]. PAs determined in the different subtypes are listed in Table 2. Only few studies were devoted to the extraction of *Borago* (see Table 3). Half of them investigated seeds or its oil [17,18,19,126], which were processed by dissolving and vortex shaking [125,126] for analysis. Only Hermann et al. refluxed the samples and isolated a glycosylated PA from *Borago* by further processing [19]. These et al. used 25% methanol in 2% formic acid for maceration and reached with their screening method based on tandem MS sensitivities in the low ppb region [125]. Despite the simpler matrix and the usage of an Orbitrap, Vacilotto et al. reached a LoQ of 0.325 ppm for the analysis of PAs in oil [125]. Dodson et al. determined the PAs in different plant parts and seeds with gas chromatography [18]. A similar experiment was done by Larson et al. investigating the PA content of fresh leaves and roots by TLC and visualizing them via Ehrlichs reagent [17].

### 5.4. Brachyglottis

With its 39 species that are all native to New Zeeland, *Brachyglottis* belongs to the aster family. It is used to treat sores and wounds [178]. Antifungal [179], antiviral [178], anti-microbial [180], and anti-cancer [178,180] properties were studied. Extraction of plant parts (listed in Table 3) was performed by maceration with ethanol [20] or methanol [21]. Inherent PAs are summarized in Table 2. Quantification was carried out via GC-MS [20,21].

### 5.5. Cineraria

*Cineraria* includes 35 species and is used in traditional medicine due to its abortive, uterus cleaning properties as well as for its potential for treating chickenpox [181]. Tunudis et al., El Shazly et al., and Wiedenfeld et al. investigated the PAs inherent to *Cineraria* species (see Table 2). They used maceration with methanol [22,24] or grinding and soaking [23]. Separation and quantification was performed by GC-MS [22,23] (see Table 4). 

### 5.6. Cynoglossum

The genus *Cynoglossum* shows strong affinity to some of its neighboring genera (e.g., *Pardoglossum* or *Solenanthus*) [182,183]. With over 75 species distributed worldwide, it is used in traditional medicine for treating illness with fever, headache and sweating [184]. Moreover, *Cynoglossum clumnae* revealed to exert cytotoxic [185,186], antimicrobial [185,186,187], antifungal [185], antidiabetic, antihyperlipidemic, and antioxidant effects [188]. Only a few PAs were identified in the genus *Cynoglossum* (see Table 2). Extraction was performed with Soxhlet extraction using methanol [25,130] or maceration with ethanol 95% [158], sulfuric acid 0.5 N [127,128], methanol [129], or hydrochloric acid 0.5 N [26]. Mroczek et al. extracted the PAs by refluxing with 1% tartaric acid in methanol [131]. No research group has to date investigated, optimized, or compared the influence of different extraction conditions on the PA content with the same plant material. The most promising extraction procedures were used by Mroczek et al. based on results achieved for comfrey [131]. Considering these results, there may be losses anticipated during long term extraction at elevated temperatures [25,127,130] because of PA degradation. Mattocks used GC-MS for identification of the different PAs in the genus *Cynoglossum* [158], while El-Shazly et al. for screening of the alkaloid profiles in stems, leaves, flowers, fruits, and roots [26]. Van Dam et al. quantified the individual PAs with GC-MS [129] before determining the sum with a photometric color-reaction based method [127,128,129] developed by Mattocks [189]. HPLC-DAD-MS was used for PA screening of different plants too [131]. A screening of different plants during maturation, without further purification, was done by Pfister et al. using q-NMR [130]. Purification or separation without direct quantification was done by TLC [25,129,158]. 

### 5.7. Eupatorium

In sum, there are over 45 different species of the genus *Eupatorium*. Different subtypes were investigated due to their pharmaceutical applications based on their antimicrobial [190,191,192], wound healing [192,193], analgesic [193], anti-inflammatory [192,193], antidiabetic [192], hepatoprotective [192], and antioxidant [192] effects. Echinatine, lycopsamine, and intermedine are the most reported PAs in *Eupatorium* (see Table 2). Samples are extracted mainly by maceration [27,29,30,31,32], percolation [33], or using Soxhlet [34] (see Table 3). Colegate et al. compared qualitatively maceration in methanol over 16 h at room temperature with hot water infusions or decoctions [31] (see Table 5). Hot water infusions and decoctions lead to high amounts of all PAs [31], whereas alcoholic tinctures mainly contain increased contents of free bases. This underlines that short extractions at high temperatures can result in increased yields of PAs. 

For the determination of PAs (see Table 4), most research groups applied GC-MS using ionization methods such as electron impact (EI) [16,30,34] or chemical Ionization (CI) [28,30] in negative or positive ion mode. Edgar et al. applied FAB (fast atom bombardment), a rarely used ionization technique in MS [28]. No data regarding the sensitivity were reported by them, thus no advantages of the different ionization techniques can be stated. Colegate et al. [31] and Kast et al. [32] used liquid chromatography with mass spectrometric determination of PAs. Kast achieved an LoQ of 1.0 × 10^−3^–3.0 × 10^−3^ ppm by using a triple quadrupole and improved sensitivity (LoD: 1.0 × 10^−3^ ppm; LoQ: 3.0 × 10^−3^ ppm) applying a high-resolution q-ToF [32]. Both groups achieved this high sensitivity due to the usage of selective detection methods, which should be considered state-of-the-art for the determination of PAs. Other research groups address analytics in a simpler way by using TLC for purification [27,29,33,34] or analysis [30] together with chloro-anillin or Mattocks reagent for visualization.

### 5.8. Heliotropium

*Heliotropium* belongs to the subgenus *Heliotropiiodeae* within the borage family. In traditional medicine, *Heliotrpium indicum* is used to treat abdominal pain, amenorrhea, dysmenorrhea, skin rashes, wounds, etc. [194,195,196]. Furthermore anti-cancer [194,195,196], anti-inflammatory [195,196], hypotensive [197], and anthelmenic [198] effects have been reported. Only a few extraction approaches (see Table 3) for PAs inherent in *Helitropium* species (see Table 2) were conducted. Methanol or either ethanol are used mostly for percolation [39,40,41,42,47,48,136] or maceration [35,36,37,38,43,45,46,132,133,134,135]. These et al. achieved, again, good results by the usage of a mixture of methanol and formic acid for maceration [125]. This procedure might be surpassed by that of Birecka et al. refluxing the samples [135,137], because, as seen before, increased temperatures have a positive effect on the yield of extracted PAs.

Most research groups were mainly interested in elucidating PA structures by applying ^1^H-NMR, ^13^C-NMR and MS [33,35,36,37,38,39,40,41,42,43,44,46,47,48,49,132,133,134,136]. If quantified (see Table 4), the focus was set on determination of PA levels in young and matured leaves [135] or in elucidating different seasonal or environmental patterns [45]. Mostly GC-MS was used [45,135,137,160]. By dividing the extracts into two fractions followed by Zn/HCl reduction of one of them, Tosun et al. quantified the PA content and the ratio of free bases to N-oxides in seeds of *Helitropium* [160]. Birecka et al. used four different methods for PA analysis. Identification was done by TLC, using Dragendorffs or Ehrlichs reagent. Quantification of several PAs was done by GC-MS, and the sum was determined using a photometric detection at 565 nm and a titration with toluene sulfonic acid [137]. These et al. applied their LC-MS/MS screening method on *Heliotropium* species [125]. However, in none of these articles are LoDs or LoQs stated. Nonetheless, it may be safely assumed that the LC-MS/MS method revealed the best selectivity with sensitivities in the low ppb region, as stated before. 

### 5.9. Lithospermum

The genus *Lithospermum* includes 59 species. *Lithospermum erythrorhizon*, as an example, is used in Traditional Chinese Medicine (TCM) to treat wounds, burns, and dermatitis. Nowadays, it is used as adjuvant for treatment of cancer [199,200]. Additionally, anti-inflammatory [199,201], antifungal [201], and antiviral [202] effects have been reported. Different PAs are reported to be inherent in the genus *Lithospermum* (see Table 2). Plant material was extracted using Soxhlet extraction with methanol over seven days [50,51,53], refluxed under acidic conditions [131]. Further a fourfold sonication-supported maceration at 40 °C was performed by Pietrosiuk et al. [52]. Based on the insights gained from previous extraction experiments losses in the PA content due to long-term extractions at increased temperature cannot be excluded. However, the two other procedures seem to result in optimal PA yields.

Regarding analytics, the focus in the literature appears to be on isolation and structure elucidation [50,51,53] rather than selective and sensitive PA determination (see Table 4). TLC was mostly used for purification, e.g., prior to GC-MS analysis [50,51]. For example, Pietrousk et al. determined shikonin derivatives inherent in hairy roots. They separated them by TLC followed by visualization with Ehrlichs reagent and GC-MS analysis [52]. Mroczek et al. repeatedly tested his HPLC-DAD-thermobeam MS method [131]. However, no quantification results are reported. 

### 5.10. Petasites

About 15 species are known to belong to the genus *Petasites* and are distributed in the northern regions in the world. Several of them are used as herbal supplement due to their antispasmodic and anti-inflammatory effects and are included in herbal medicinal products as agent for treatment of spasms of urogenital tract, gastrointestinal colic and dysmenorrhea [203]. It is currently investigated in clinical trials for its effectiveness in migraine prophylaxis [204]. For *Petasites,* different inherent PAs have been reported (see Table 2). They were extracted using maceration [55,57,58,59], refluxing [54] or Soxhlet [55] (see Table 3). Schenk et al. analyzed a CO_2_-Extract [138] and Knez et al. developed a procedure with propane (HPPE, high pressure propane extraction) [139] for large scale extraction of PAs. Niwa et al. showed some evaluation of the PA extraction efficiency (see Table 6) [55]. They extracted *P. japonicus* samples on the one hand with boiling water and on the other hand with ethanol by Soxhlet for 24 h. They stated that only half of the PAs (54%) could be extracted with hot boiled water compared to the ethanolic extraction procedure. However, due to the different solvents, time, and procedures applied, it is difficult to compare the results. It can be suggested that an increased PA content is achieved if ethanol is used at increased temperatures for elongated extraction times. 

Considering analytics of *Petasites* (see Table 4), Wildi et al. developed a TLC method with UV detection accomplishing a LoD of 1 ppm, which can compete with HPLC- and GC-based techniques [57]. PAs in the genus *Petasites* were analyzed in most cases by liquid chromatography. Schenk et al. used a coupling of UHPLC with ToF-MS [138] and achieved a quantification limit of 0.002 ppm (LoD not given) for *Petasites.* That sensitivity limit was not reached by combination of ToF-MS with conventional HPLC separation (LoD: 0.01 ppm; LoQ: 0.5 ppm) [56,205]. One explanation for the higher sensitivity is that peaks achieved by UHPLC are narrower and therefore higher due to smaller particles and higher pressure used. Satisfying results covering a LoQ from 0.35 to 25.0 ppm were also achieved by using UV detection. Because of missing chromophores, Niwa et al. determined PAs with a RI (refractive index) detector [55]. The smallest calibration point was set at 10.0 ppm, leaving thus the question regarding the LoQ unanswered. Selective and sensitive quantification of PAs without prior chromatographic separation was achieved by using an enzyme linked immunosorbent assay (ELISA) specific for retrorsine [57,58,59]. Interesting results were achieved by Langer et al. developing antibodies against retrorsine for the determination of senecionine; selectivity was shown and a LoD of 0.1 ppm in plant material was reached [58]. A disadvantage is that only one PA is determined. 

### 5.11. Senecio

More than 1000 subspecies of the genus *Senecio* are distributed worldwide. Therefore, several of them were investigated for their medicinal effects. For example, *Senecio scandens* Buch.-Ham. ExD. Don is exported as OTC in traditional Chinese medicine for treating bacterial diarrhea, enteritis, conjunctivitis, and respiratory tract infections [206] or in Africa against malaria [207] or hypertension [208]. Other species were tested for their relaxant effects [209] as well as anti-inflammatory [155,206,209], antimicrobial [155,206], antileptospirosis [206], hepaprotective [206], anti-infusorial [206], antioxidant [206,210,211,212,213,214], antiviral [206,215,216], antitumoral [155,206,217], analgesic [206], anti-tuberculosis [218], anti-spastic [155], and antibacterial [103] characteristics. In several *Senecio* species, 39 different PAs have been identified (see Table 2).

Maceration under acidic conditions was mostly applied, e.g., 0.05 M HCl [65,104,140], 0.1 M HCl [92], 1 M HCl [68], 0.05 M H_2_SO_4_ [72], 0.1 M H_2_SO_4_ [63,95], or 0.4 M formic acid [66], as well as methanol [64,75,85,97,106,142,143,219,220], ethanol [79,84,109,144,146], or other organic mixtures [73,98]. A large scale maceration apparatus, based on acidic solvents, was developed by Craig et al. [221]. Alternative techniques such as ultrasonic [65], supercritical fluid (SFE) [78], continuous [101], and cold ion extraction [81] were performed as well. Extractions at increased temperatures were mostly performed by Soxhlet [14,74,76,77,82,83,91,96,99,103,108,145,147,148,149,150,151,165] or refluxing [25,88,105] with the extraction of *Senecio* studied by a few research groups in more detail.

Hartmann et al. compared the spontaneous reduction of N-oxides if Soxhlet extraction or maceration was applied. They showed that up to 50% of the N-oxides were reduced during prolonged Soxhlet extraction [72] due to thermal degradation (see Table 7).

Bicchi et al. varied pressure and temperature of SFE, and compared the results with Soxhlet extraction (see Table 8) [78]. It can be concluded that both temperature (Experiment 2 vs. 4) and pressure (Experiment 3 vs. 4) have a significant influence on the extraction yield. However, this study does not specify as to which parameter is more important. This may therefore constitute additive effects, as despite the higher temperature applied during Soxhlet extraction SFE revealed higher contents. This underlines the importance of increased pressures for an exhaustive extraction. The evaluation of the ideal Soxhlet conditions was done by varying the sample amount of *S. inaequidens* between 1 and 25 g and the extraction time between 0.5 and 8 h. As a result, they limited the extraction time to 4 h [76], which may be a further evidence for thermal degradation. 

Kopp et al. [122] investigated the influence of solvent (acidic or basic), concentration (1% or 5%), temperature (25–125 °C), and pressure (see Table 9) using PLE. The results were compared with a BfR-based maceration method [222]. It was shown that solvent is one of the most important parameters. Temperature dependent differences of up to 600 ppm in final PA content were shown. No effect of temperature was observed between 50 and 75 °C, but, if increased above 100 °C, dependent on solvent, a loss of PAs could be detected. This loss can be correlated with PA degradation caused by reactive side chains. Comparing these PA contents with the maceration-based method revealed higher PA yields of up to 380 ppm at moderately increased temperature and increased pressure. The direct influence of pressure could not be clarified during this study. 

Zhang et al. compared refluxing (methanol), soaking or sonication assisted extraction of samples in aqueous HCl [104]. They decided to use sonication experiments for optimizing adonifoline content, because of the easy, fast, and rugged handling without thermal degradation. A L9 (34) orthogonal study design (also referred to as the Taguchi design) (see Table 10) was used.

Varied parameters were solvent volume, concentration, and extraction time. Zhang et al. concluded that, in the case of adonifoline, the most important factor appears to be time followed by concentration and volume. Hence, the best results were achieved if low concentrations of hydrochloric acid were applied for a long time, i.e., 40 min. 

To summarize the results for the extraction of plants of the genus *Senecio*, it may concluded that best yields can be achieved by applying acidic solvents at increased temperatures for short time periods preferably at high pressures. 

Analyzing PAs in the genus *Senecio* (see Table 4), the whole bandwidth of chromatographic techniques were used. Unfortunately, the achievable sensitivity is rarely reported. For example, GC coupled with FID [68,72,77,78,82,84,87,95], NPD [68,72,87], or MS [62,63,67,68,73,74,75,76,77,83,84,87,92,94,99,103] were used but no LoDs or LoQs are stated. The same holds true for HPLC-based analyses. Only a few research groups published their achieved sensitivities after optimization. Zhang et al. reported via MS/MS detection based methods a LoD of 0.5 × 10^−3^ ppm [104]. A comparable result was achieved for adinofoline in UV, after previous derivatization with chloro-aniline [140]. Schaneberg et al. reported only medium sensitivity when ELSD was coupled to HPLC [64]. The best LoD (0.3–1.1 × 10^−6^ ppm) was achieved by Zhou et al. with UHPLC combining high resolution with short analysis times and reduced solvent consumption [65]. Determination without previous separation was mainly done by quantitative NMR (q-NMR) and enzyme linked immunosorbent assay (ELISA). Pieters et al. compared q-NMR (^1^H and ^13^C) with GC and concluded that there is a lack of selectivity due to the complex spectrum with much overlapping signals using ^1^H-NMR. In contrast, ^13^C-NMR might be more suitable, but it cannot compete with the sensitivity achieved by GC-MS [80]. Langer et al. developed specific antibodies against retrorsine and immobilized them for the determination of senecionine, the main alkaloid of genus *Senecio*. Despite an LoD of 0.023 × 10^−3^ ppm and the resolved cross-reactivities against monocrotaline, senecyphilline, retrorsine-N-oxide, and senkirkine, its usage is limited to a single PA [58].

### 5.12. Symphytum

*Comfrey* is used as a medicinal plant based on its anti-inflammatory and analgesic effects. Multiple trials have demonstrated the efficacy of comfrey preparations for treatment of pain, inflammation, and swelling of muscles in degenerative arthritis, sprains, acute myalgia in the back, contusions, and strains after sport injuries [223]. Several PAs were identified in roots or leaves of *Symphytum* (see Table 2). The dried, freeze-dried [153], or nitrogen-ground [224] plant parts were extracted by various extraction procedures and solvents (see Table 3). In most cases, maceration with or without sonication was utilized for PA extraction [64,93,116,152,153,154,155]. Other procedures such as refluxing [54,116,152], percolating [152], HPWE [116] or PLE [122] were used too. Only limited studies were devoted to investigate the effect of different extraction conditions on the yield of PAs from *Symphytum*. The most detailed study was presented by Mroczek et al. investigating the effect of solvents (organic, aqueous, acidified, alkalified, or mixtures), temperature, time, or technique (maceration, percolation, sonication, or refluxing) [152] (see Table 11).

The significant influence of the solvent could be shown by comparing the results of a 1% methanolic solution of tartaric acid (highest yields) with methanol (second highest yield), 2.5% aqueous hydrochloric acid (third highest yield), followed by ethanol 95% and an alkalified mixture of methanol and chloroform. However, it was also shown that this effect can be eliminated at elevated temperatures. However, increased temperatures and selecting the least efficient solvent may result in strongly decreased PA amounts. For example, high temperatures in combination with hydrochloric or ascorbic acid resulted in decreased amounts. Considering time, extended extraction periods at room temperature result in higher yields, yet in losses at higher temperatures (4-h reflux Experiment 12 compared to 2-h reflux experiment), which may come from degradation. Overall, experiments done by Mroczek et al. show that contents between 163 and 1300 ppm were reached for the same plant material dependent on the selected extraction parameters. In summary, acidic solvents and increased temperatures for short time periods appear to be the parameters of choice.

Feng Liu et al. extracted samples by sonication, refluxing and pressurized hot water extraction. Solvent (methanol, ethanol, and mixtures), time, and temperature were varied (see Table 12). The 6 ppm higher content using water/methanol for extraction reflects that solvents have an important influence. In combination with an increased temperature (i.e., refluxing), a fourfold content of lycopsamin was achieved, which reveals the additional positive effect of increased temperatures. Even PHWE lead to higher lycopsamine contents compared to those obtained with sonication [116]. 

Kopp et al. studied the extraction with PLE [122] and compared the results with the BfR-based extraction method [222]. Solvent, concentration, and temperature were varied systematically, while pressure was held constant during PLE experiments (see Table 13).

Due to the inherent mucilage, *comfrey* was a challenging matrix resulting in cell clogging during extraction. Therefore, ammonia and acetic acid were not considered. Best yields were shown for PLE extraction on increased temperatures with strong acids at high concentration levels (e.g., sulfuric acid). The evaluations showed that solvent was the most important factor, but selecting a less ideal solvent composition may be compensated by increasing the extraction temperature, as shown by Mroczek et al. [152]. Comparison with the maceration method reveal the positive influence of temperature and pressure, because a threefold higher content of PAs was found in PLE extracts.

On the analysis, Roeder et al. [169] and Mroczek et al. [152] used a photometric quantification method for determining the sum of PAs. A LoD of 1 ppm was reported by Roeder for this method. In the case of *comfrey*, which has high contents of PAs, this may be acceptable. If smaller concentrations are apparent, matrix effects due to accompanying substances have to be considered. Most research groups used methods such as HPLC and GC for quantification, and TLC and CCC (counter current chromatography) for preparative purification [113]. Janes et al. used TLC for analytical purposes and developed a screening method with densiometric detection (LoD: 22 ppm) [168]. Schaneberg et al. improved selectivity using LC-ELSD (Evaporating Light Scattering Detection), but only reached a LoD of 40 ppm [64]. Better results were achieved by Mroczek et al. [54], who developed a sensitive HPLC- DAD method with a LoD of 0.06 ppm. Determinations of PAs by GC, to enhance sensitivity, were made mainly by MS [114,153] and FID [93,114] as detection method. No LoDs have been reported for these methods in the literature. Similar sensitivity to MS/MS was achieved by Oberlies et al. using GC coupled with a nitrogen selective detector (NPD) [154]. Other methods have an LoD of approximately 1 × 10^−3^ ppm. The sensitivity can be improved by using an orbitrap that enables MS^n^, but no LoD was stated by Liu et al. [116]. 

### 5.13. Tussilago

Herbal preparations of *Tussilago* were used in traditional medicine for treating cough, phlegm, bronchitic, and asthmatic conditions [225]. Today, different articles exist that prove the antimicrobial [226,227], antitubercular [228], antioxidant [229,230], neuroprotective [229], antitussive [225], expectorant [225], and anti-inflammatory [225] effects. Senkirkin is the most commonly reported PA detected in *Tussilago* (see Table 2). The different techniques and solvents used for extraction are listed in Table 3. Maceration of the dried plant [118,119,121,156] was found to be the preferred procedure. Only a few research groups investigated the extraction behavior of inherent PAs.

Lebada et al. studied the influence of solvent, time, temperature, plant-to-solvent ratio, and the extraction procedure applied on the extraction efficiency of senkirkine [118] (see Table 14).

It was derived that a low plant-to-solvent ratio of 1 g/60 mL and 1 g/100 mL results in a 3 ppm higher yields than shown for a ratio of 1 g/30 mL. Regarding extraction solvent, best yields were achieved by using a mixture of methanol/water (1:1) acidified with citric acid. Alkaline solvents result in a decreased senkirkine content. The influence of temperature was more drastic, as a threefold higher PA content could be obtained for samples extracted with water, when increased temperature was applied for a short time period. The opposite was shown for high temperatures over a long time period. In the case of Soxhlet extractions lasting for 48 h, a loss of 8.7 ppm senkirkine was noted compared to the best extraction.

Jiang et al. compared Microwave Assisted Extraction (MAE), Pressurized Hot Water Extraction (PHWE), and heating under reflux [157] to show that PHWE is suitable for extraction of PAs from *Tussliago* (see Table 15) After optimization, PAs extracted using MAE and PHWE were comparable to that by heating under reflux. Again, a mixture of methanol/water acidified with HCl was found to be the best extraction solvent. Varying the temperature in the range of 60–120 °C did not affect the PA content significantly. It may also be concluded from this work, that using acidified solvents at increased temperatures over short time periods results in good extraction yields. 

Kopp et al. addressed the influence of temperature, kind of solvent (weak acid, strong acid, or base), concentration, and pressure applying pressurized liquid extraction (PLE) [122]. These results were compared with the BfR-based liquid–liquid extraction method [222]. Acidic extraction resulted in a threefold higher PA content compared to alkaline ones (see Table 16). Comparison of PLE yields with those of the BfR-based method underlines the assumption that the applied pressure increases the extraction rates significantly.

Summarizing the insights gained from previous results, it may be concluded that PAs are extracted best if acidic conditions at elevated temperatures and increased pressure are applied. However, sizeable extraction losses were recognized in case of long-term procedures such Soxhlet over 48 h [118]. 

Analytical techniques used for determining PA content in *Tussilago* extracts are summarized in Table 4. Culvenor et al. determined the sum of PAs of the crude extract by titration with toluene sulfonic acid [156]. Because of the lack of selectivity and sensitivity, methods with previous separation such as TLC or HPLC, based on polar, apolar, ionic, chiral, and similar interactions are used more often. Samples could be applied without cleanup or derivatization procedures and were determined by various detection methods. Mass spectrometry was preferred one because of its performance regarding sensitivity and selectivity [117,157]. Jiang et al. achieved an LoD of 0.000275 ppm for senkirkine using LC-q-ToF [157]. GC-MS was used by Nedelcheva et al. to investigate the PA distribution of the genus *Tussilago* native in Bulgaria [120].

Advantageous is the increased selectivity because of the high resolution of GC vs. liquid chromatography. Sample preparation is more time consuming because of the needed reduction step (Zn/HCl or Serdoxit columns) for the non-volatile N-Oxides. Capillary electrophoresis (CE) provides a still better resolution power than GC. It was used by Lebada et al. to quantify the content of senkirkine and senecionine. Despite missing chromophores, with LC-Q-ToF a comparable LoD was achieved. This may be attributed to the very narrow peak shapes [118]. Cao et al. used amplified sample stacking, an online preconcentration step in combination with electrokinetic chromatography (MEKC). In this combination of chromatography and electrophoresis, PAs are separated by differential partitioning between micelles and the aqueous surrounding buffer [121].

## 6. Discussion

### 6.1. Extraction Techniques

The main aim of every extraction strategy is achieving a recovery of 100% of the analyte from the respective matrix. To assess the efficiency of various PA extraction strategies from different plant parts, the literature on thirteen relevant plant genera was reviewed regarding the selected extraction conditions. A multitude of extraction procedures and solvents were applied. Plant parts were used fresh, freeze dried, dried in oven, or frozen in liquid nitrogen. Afterwards, they were crushed, and then ground or milled to obtain a powder suitable for efficient extraction. Maceration of plant material was performed in organic, aqueous, acidic, or basic solvents, or even in mixtures thereof. Samples were extracted once or multiple times, with or without treating samples at elevated temperatures, and for time periods ranging from 30 min up to a few days. The same parameters (i.e., solvent, temperature, and time) were also varied for all other reported extraction procedures. Only few research reports discuss novel extraction routines such as MAE, HWPE, SFE, or PLE, whereby the pressure is an additional parameter affecting efficient extraction. As the PA content in plant matter differs with regard to subspecies, habitat of growth, soil of growth, nutrition, and/or water supply, only relative comparisons of the results obtained by different research groups are possible. Every plant collected or grown for a PA study provides an individual PA pattern and concentration level, which renders absolute comparison of the applied techniques only feasible, if the bulk of the plant material was identical. Only a few research groups have considered this aspect so far and have thus compared various parameters on a fair basis for determining optimum conditions for close to ideal PA recoveries. Suitable comparisons of the extraction procedures using essentially the same bulk material was available for four of the thirteen plant genera reviewed herein. Evaluating these studies reveals that the content of PAs varies up to 1100 ppm [0.11% (*w*/*w*)] for a specific plant material, if different extraction methods, solvents, temperatures, and extraction durations were used.

In summary, it was shown that applying acidified mixtures of organic solvents at temperatures up to 100 °C for short periods of time apparently leads to optimum yields of PAs. The precise extraction conditions evidently depend on the character of the matrices and the molecular structures of the inherent pyrrolizidine alkaloids. If these aspects are considered, clustering of plant matrices is possible to extrapolate extraction conditions for a new plant material. It has to be considered that matrices with inherently robust PAs are best extracted at high temperatures with solvent mixtures containing strong acids. Comparatively simple matrices with PAs comprising sensitive functional groups are best extracted using solvent mixtures containing weak acids at slightly increased temperatures. However, for all matrices and PA properties, it was shown that short extraction times are preferred, as extractions during extended time periods resulted in significantly decreased PA recoveries.

### 6.2. Analytical Techniques

For the purpose of determining toxic substances in plants or plant extracts used for herbal preparations, appropriately sensitive and selective analytical methods are needed. Recapitulating the analytical techniques reported in literature and summarized herein, a wide range of methods were applied predominantly based on some form of separation technique. The determination of alkaloids as sum parameter without previous separation (i.e., photometrically, by titration, or by quantitative NMR using spin of ^1^H or ^13^C) remains less pronounced and appears challenging due to potential interferences resulting from accompanying substances within these extracts. Therefore, chromatographic separations appear mandatory for ensuring sufficient selectivity and maximizing the sensitivity of the coupled detection technique. For semi-quantitative or less sensitive PA determination, thin layer chromatography was successfully applied. For visualization, Ehrlichs reagent, chloro-aniline, Mattocks procedure, or Dragendorffs reagent is used in combination with densiometric detection or detection under UV radiation. More recently, an increased usage of high- or ultra-high-performance liquid chromatography has been noticed during reviewing the pertinent literature providing higher chromatographic resolution, which was particularly successful for separating pyrrolizidine alkaloids from their N-oxides. 

Due to missing chromophores, a variety of alternative detection methods including ELSD, RI, etc. were tested, yet have not achieved useful LoDs and LoQs. Hence, predominantly mass spectrometric detection techniques have been reported for achieving improved detection limits and taking advantage of the mass selectivity via quadrupole, triple quadrupole, ToF or Orbitrap detectors combined with a variety of different ionization techniques. While GC and CE enabled efficient separation of species, both appear restricted in sensitivity using UV or FID, respectively. In contrast, GC coupled with MS or NPD yielded satisfying detection limits. The main remaining disadvantage is the fact that N-oxides are not sufficiently volatile, and have to be reduced prior to the analysis (i.e., via Zn powder in acidic solution, or by applying serdoxit Serdoxit columns). Finally, CE and MEKC methods were rarely applied to date in PA analysis. Concluding, as the current state-of-the-art in terms of both selectivity and sensitivity addressing these complex matrices with sufficient analytical quality, LC-MS/MS appears the recommended technique.

## 7. Conclusions

The phytopharmaceutical industry has been analyzing pyrrolizidine alkaloids in plant extracts for almost 30 years. To date, sensitive and selective analytical methods for determination of PAs in plant extracts have been developed and applied. Surprisingly, only few research groups dedicated efforts to the development of innovative, more efficient, and exhaustive extraction strategies. Hence, to date, usually the simplest or most readily established techniques for the respective plant are applied, albeit more efficient methods may be available. Only four out of thirteen plants were thoroughly investigated with regard to optimal extraction conditions or procedures. Even less attention has been paid to the fact that the trueness of the values reported during the analytical assessment of the PA content undoubtedly hinges on the quantitative extraction of PAs for avoiding that erroneous levels are reported.

In summary, this review shows that the extraction and analytical assessment of PAs in plant extracts has a long history, yet would benefit from further evolving strategies on both innovative efficient extraction schemes and advanced molecularly selective analytical quantification techniques ensuring the safety of herbal substances and herbal extracts.

## Figures and Tables

**Figure 1 toxins-12-00320-f001:**
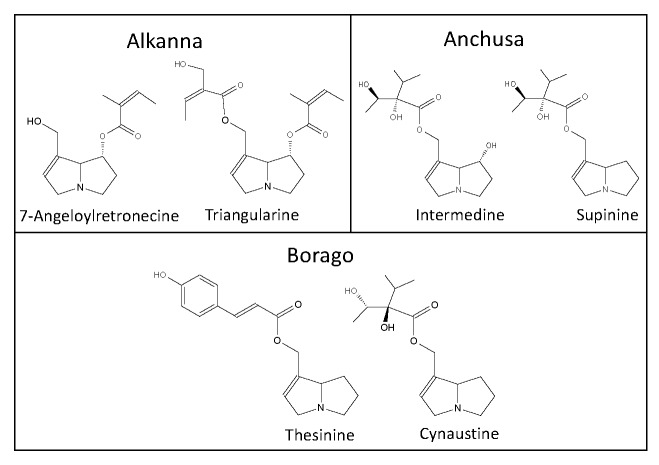
Pyrrolizidine alkaloids inherent in the genera of *Alkanna, Anchusa,* and *Borago*.

**Figure 2 toxins-12-00320-f002:**
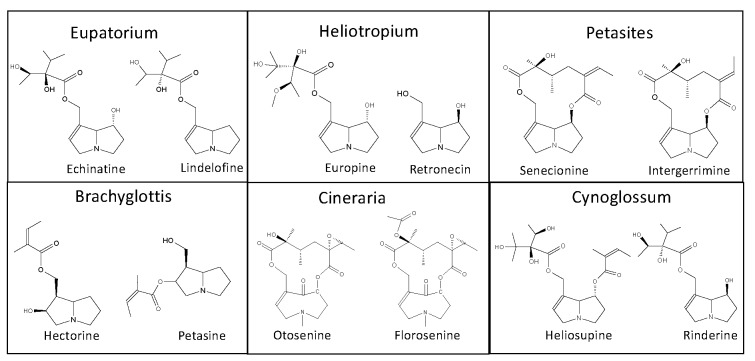
Examples for PAs inherent in plants of the genera Eupatorium, Heliotropium, Petasites, Brachyglottis, Cineraria, and Cynoglossum.

**Figure 3 toxins-12-00320-f003:**
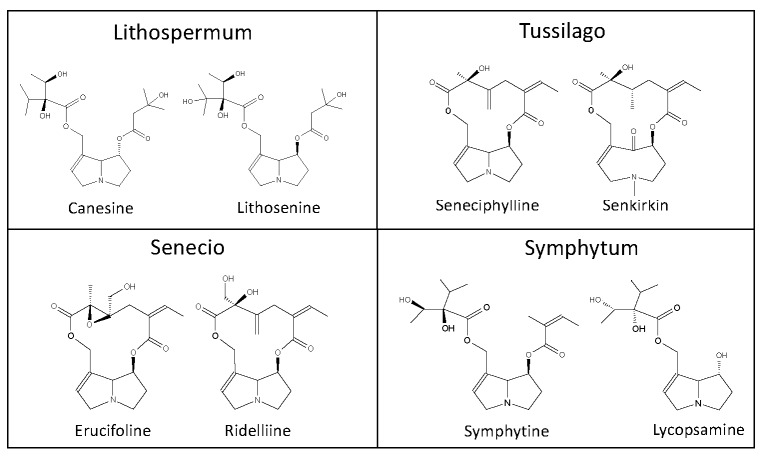
Pyrrolizidine alkaloids inherent in plants of the genera *Lithospermum, Tussilago, Senecio*, and *Symphytum.*

**Table 1 toxins-12-00320-t001:** Concise description of extraction techniques used. All technical variations are based on maceration or percolation.

Technique	Temperature	Pressure	Sample	Solvent	Remarks
Maceration	≤boiling point	atmospheric	e.g., flask	any solvent	sample is extracted by soaking in solvent
Refluxing	≤boiling point	atmospheric	round bottomed flask	any solvent	maceration on increased temperature, vaporization of solvent is avoided by condensation
Soxhlet	≤boiling point	atmospheric	Soxhlet cartridge	any solvent	special form of percolation (continuous)
Percolation	room temperature	atmospheric	e.g., column	any solvent	sample placed in column, solvent is added, flow through and is released
Sonication	room temperature	atmospheric	e.g., flask	any solvent	maceration assisted by sonication to increase solubility
SFE (Supercritical Fluid Extraction)	>boiling point	pressurized	reaction vessel	e.g., CO_2_	temperature and pressure above critical point to control extraction characteristics
PLE (Pressurized Liquid Extraction)	>boiling point	pressurized	reaction vessel	no corrosive/decomposing solvents	pressurization allows temperatures above boiling point, faster extractions
MAE (Microwave Assisted Extraction)	>boiling point	atmospheric	reaction vessel	no decomposing solvents	pressurization or pressure stabile reaction vessels allow temperatures above boiling point, faster extractions heated by radiation
HPPE (High Pressure Propane Extraction)	>boiling point	pressurized	reaction vessel	propane	variation of SFE
Cold Ion Exchange	room temperature	atmospheric	column	any solvent	plant material placed in column solvent is pumped continues in cycle, analyte is adsorbed on specific material
PHWE (Pressurized Hot Water Extraction)	>boiling point	pressurized	column	water (different modifier)	sample is placed in column, hot water (modifier) is pumped through column

**Table 2 toxins-12-00320-t002:** Overview of PAs detected and identified in different genera. No distinction between N-oxides and free bases is made, as this was not differentiated by all research groups. The main name describes the PA and its derivatives.

Genus	Identified PAs
*Alkanna* (Boraginaceae)	7-Angeloylretronecine [14,15], 9-Angeloylretronecine [14], 7-Tigloylretronecine [14], 7-Senecioylretronecine [14], 9-Tigloylretronecine [14], 9-Senecioylretronecine [14], 7-Angeloyol-9-(hydroxypropenoyl) retronecine [14], 7-Tigloyl-9-(hydroxy propenoyl) retronecine [14], 7-Angeloyol-9-(2,3-dihydroxypropanoyl) retronecine [14], 7-Tigloyl-9-(2,3-dihydroxypropanoyl) retronecine [14], Triangularine [14,15], Triangularicine [14], Dihydroxytriangularicine [14,15]
*Anchusa* (Boraginaceae)	Anthamidin [14], Supinine [14,16], Intermedin [14,16], Lycopsamine [14,16], Currassavine [16]
*Borago* (Boraginaceae)	Lycopsamine [17], Supinidine [17],Viridiflorate [17], Cynaustine [17], Amabaline [17,18], Thesinine [18,19]
*Brachyglottis* (Asteraceae)	Senecionine [20,21], Retrorsine [20,21], Integerrimine [20], Senkirkine [20], Hectorine [20,21], Petasin [20,21], Angeloylheliotridine [20], Clivorine [20]
*Cineraria* (Asteraceae)	Otosenine [22,23], Florosenine [22,23], Floridanine [22,23], Doronine [22], Senecionine [23,24], Integerrimine [23,24], Seneciphylline [23,24], Jacobine [23,24], Usaramine [23]
*Cynoglossum* (Boraginaceae)	Heliosupine [25,26], Rinderine [25,26] Echinatine [25,26], Viridiflorine [26]
*Eupatorium* (Asteraceae)	Lindelofine [27], Supinine [27,28,29,30], Lycopsamine [30,31,32], Intermedin [30,31,32], Amabaline [28,33], Echinatine [28,29,30,32,33], Rinderine [28], Viridiflorine [28], Cynaustraline [28], Tussilagine [34]
*Heliotropium* (Boraginaceae)	Trachelanthamine [35,36], Floridine [35], Heliovicine [35], Lycopsamine [37,38], Intermedin [38], Amabiline [37], Curassavine [37,39], Heliospathine [37], Europine [40,41,42,43], Liamin [40], Heliotrine [42,43,44,45], Lasiocarpine [42,45,46], Retronecine [38,47,48,49], Helibracteatine [47], Helifoline [48], Heliscabine [49], Heliosupine [44,46], Echinatine [44], Supinine [43], Heleurine [43], Coromandaline [39]
*Lithospermum* (Boraginaceae)	Lithosenine [50], Lycopsamine [51], Canescine [51,52] derivatives, Canescinine [51], Intermedine [51,53], Mysocorpine [53]
*Petasites* (Asteraceae)	Senkirkine [54,55,56], Senecionine [54,56,57,58,59], Intergerrimine [58,59], Petasitenine [55], Neopetasitenine [55]
*Senecio* (Asteraceae)	Ridelline [60,61,62,63,64,65,66], Retrorsine [60,61,62,63,64,65,66,67,68,69,70,71,72,73,74,75,76,77,78,79,80,81,82,83,84,85], Seneciphylline [60,61,62,63,64,65,66,67,68,70,71,72,75,77,78,80,81,82,83,84,86,87,88,89,90,91,92,93,94,95,96,97,98,99], Senecionine [60,61,62,63,64,65,66,67,68,69,70,71,73,74,75,76,77,78,79,80,81,82,83,84,86,87,89,90,91,92,93,94,95,96,97,98,99,100,101],Senkirkine [60,61,62,65,68,73,74,77,79,80,83,89,97,98,101], Jacobine [68,78,82,87,91,94,95,96,99], Integerrimine [61,62,63,64,66,68,69,70,71,72,73,74,75,76,77,78,79,80,82,83,84,87,89,92,93,94,95,97,99,100,101], Spartiodine [61,62,63,66,68,70,77,78,82,90,92], Senecivernine [62,63,66,68,69,71,73,74,75,76,77,78,80,82,87,94,97], Platyphylline [61,62,100,102,103], Usaramine [61,63,66,68,69,70,71,73,74,77,78,79], Adinofoline [61,68,104], Florosenine [68,71,73,74,77,94,98,101], Erucifoline [68,83,87,94,95], Otosenine [68,71,73,74,77,88,94,97,101], Triangularine [75,100,105], Sarracine [75,100,105], Sarracinine [100], Eruciflorine [68,87], Onetine [68], Floridanine [68], Senecicannabine [68,106], 7-Angeloylheliotridine [97,103], 9-Tiglylplatynecine [75,100], 7-Angeloylretronecine [75,97], Petasin [97], 9-Angeloylplatynecine [75], Monocrotaline [65], Uspallatine [69,85], Rosmarinine [90,107,108], Angularine [107,108], Hadiensiene [107], Ruwenine [109], Ruzorine [109], Doriasenine [110], Sceleratine [111]
*Symphytum* (Boraginaceae)	Echimidine [112,113,114], Symphytine [112,113], Lasiocarpine [64], Intermedin [115], Lycopsamine [114,115,116]
*Tussilago* (Asteraceae)	Senkirkine [117,118,119,120], Senecionine [116,118,119,120,121], Intergerrimine [120], Seneciphylline [120], Senecivernine [122]

**Table 3 toxins-12-00320-t003:** Overview of the extraction techniques and solvents used to process plant extracts.

Genus	Technique	Solvent
*Alkanna* (Boraginaceae)	Maceration	0.5 N HCl [14], Methanol [15]
*Anchusa* (Boraginaceae)	Maceration	0.5 N HCl [14], Chloroform [123,124], Methanol [123,124], Methanol/Water/Formic Acid (25/2/73) [125]
*Borago* (Boraginaceae)	Maceration	Methanol/Water/Formic Acid (25/2/73) [125], Methanol/Water [126], Hexane [18]
	Refluxing	Methanol/Water (4:1) [19]
*Brachyglottis* (Asteraceae)	Maceration	Methanol [21], Ethanol [20]
*Cineraria* (Asteraceae)	Maceration	Methanol [22,24], 0.5 N HCl [23]
*Cynoglossum* (Boraginaceae)	Maceration	0.5 M Sulfuric Acid [127,128], Methanol [129], 0.5M HCl [26]
	Soxhlet	Methanol [25,130]
	Refluxing	Tartaric Acid in Methanol [131]
*Eupatorium* (Asteraceae)	Maceration	Methanol [27,29,30,31], Water [31], 0.05 M Sulfuric Acid [32]
	Soxhlet	Methanol/Dichlormethane [34]
	Percolation	Ethanol [33]
*Heliotropium* (Boraginaceae)	Maceration	Methanol [36,37,39,45,46,132], Ethanol [35,38,43,133,134,135], Methanol/Water/Formic Acid (25/23/2) [125]
	Percolation	Methanol [40,41,42], Ethanol [39,47,49,136]
	Refluxing	Methanol [137]
*Lithospermum* (Boraginaceae)	Soxhlet	Methanol [50,51,53]
	Refluxing	Tartaric Acid in Methanol [131]
	Sonication	Methanol [52]
*Petasites* (Asteraceae)	Maceration	Methanol [57,59,93], Water [55]
	Refluxing	Methanol/Tartaric Acid [54]
	Soxhlet	Ethanol [55]
	SFE	CO_2_ [138]
	HP-Propan	Propane [139]
*Senecio* (Asteraceae)	Maceration	0.05 M Sulfuric Acid [72], 1 M Sulfuric Acid [63], 0.05 M Hydrochloric Acid [140], 0.1 M Hydrochloric Acid [92], 1 M Hydrochloric Acid [68], Chloroform 0.1 M Hydrochloric Acid (1:1) [73], Methanol [64,69,75,85,90,97,106,140,141,142,143], Ethanol [79,84,107,144,145,146], 0.4 M Formic Acid [66], Et2O/Pertolum Ether/Methanol (1:1:1) [98], 0.05 M Sulfuric Acid [72]
	Refluxing	Methanol [86,91,93], Ethanol [61]
	Soxhlet	Methanol [71,74,76,82,83,86,91,96,99,100,103,107,108,147,148,149,150,151]
	Sonication	0.05 M Hydrochloric Acid
	SFE	CO_2_ [78]
	PLE	Sulfuric Acid [122], Phosphoric Acid [122], Ammonia [122], Acetic Acid [122], Formic Acid [122]
	Cold Ion Exchange/ Continuous Extractor	Methanol [81], Petroleum Ether [101]
*Symphytum* (Boraginaceae)	Maceration	Methanol (hot) [93], Methanol [64,152], 0.025 M Sulfuric Acid [153], Water (hot) [154], Tartaric acid in Methanol [152]; Ethanol [152] 0.7 M HCl [152], Chloroform/Methanol [152]
	Refluxing	Tartaric Acid in Methanol [54,152], Methanol/Water (50/50) [116], Ethanol [152], 0.7 M HCl [152], Ascorbic Acid in Methanol [152]
	Sonication	Chloroform (basic) [155], Methanol/Water (50/50) [116], Methanol/Chloroform (15/85) [116], Methanol [116], Ethanol [116], Tartaric Acid in Methanol [152], Acetic Acid [152]
	Percolation	Methanol [152]
	HWPE	Water [116]
	PLE	Acetic Acid [122], Phosphoric Acid [122], Formic Acid [122], Sulfuric Acid [122], Ammonia [122]
*Tussilago* (Asteraceae)	Maceration	Methanol/Citric Acid [118,119,121], Methanol/Ammonia [118], 0.25 M Sulfuric Acid [156], Water [118], Acidified Water [118]
	Microwave	Methanol/Water acidified with hydrochloric acid or acetic acid [157]
	Refluxing	Methanol/Tartaric acid [117], Water [54,118], Methanol alkaline [118], 1 M HCl [157]
	Soxhlet	Methanol [118]
	PHWE	Hot water [157]
	PLE	Acetic acid [122], Phosphoric acid [122], Formic acid [122], Sulfuric acid [122], Ammonia [122]

**Table 4 toxins-12-00320-t004:** Overview of analytical techniques and detection methods for the determination of PAs in plant extracts. The LoD or LoQ is listed if given in corresponding manuscript, if not it is characterized by “--”.

Genus	Separation	Detection	LoD (ppm)	LoQ (ppm)
*Alkanna* (Boraginaceae)	GC	MS [14]	--	--
	DCCC	UV [15]	--	--
*Anchusa* (Boraginaceae)	GC	MS [14,16]	--	--
	HPLC	MS [124,125]	--	--
*Borago* (Boraginaceae)	GC	MS [18,19]	--	--
	HPLC	MS [126]	--	--
	HPLC	MS/MS [125]	--	--
	HPLC	Orbitrap [125,126]	--	0.325
	TLC	Ehrlichs Reagent [17]		
*Brachyglottis* (Asteraceae)	GC	MS [20,21]	--	--
	TLC	Visual [20,21]	--	--
*Cineraria* (Asteraceae)	GC	MS [14,22]	--	--
*Cynoglossum* (Boraginaceae)	HPLC	MS [25,131]	--	--
	GC	MS [26,129,158]	--	--
	TLC	Visual [25,158]	--	--
	none	Photometric [127,128,129]	--	--
	none	q-NMR [130]	--	--
*Eupatorium* (Asteraceae)	HPLC	MS/MS [31,32]	--	--
	GC	MS [28,30,34,159]	--	--
	TLC	Visual [27,29,33], Chloranillin [30], Mattocks Reagent [34]	--	--
*Heliotropium* (Boraginaceae)	HPLC	MS/MS [125]	--	--
	GC	MS [45,135,137,160]	--	--
	None	p-Toluene Sulfonic Acid [137]	--	--
	TLC	Dragendorffs [137]	--	--
	None	Photometric [137]	--	--
*Lithospermum* (Boraginaceae)	GC	MS [50,51,52,131]	--	--
	HPLC	DAD [131]	--	--
	HPLC	MS [131]	--	--
	TLC	Ehrlichs—Reagent [52]	--	--
*Petasites* (Asteraceae)	HPLC	UV [54,55,56,161]	0.10–5.0	0.35–25.0
	HPLC	RI [55]	--	<10
	HPLC	ToF-MS [56,161]	0.01	0.50
	UPLC	ToF-MS [138]	--	0.002
	GC	FID [93]	2	--
	TLC	Densiometric [54]	20	40
	TLC	UV [57]	1	--
	None	Photometric [162]	--	--
	None	ELISA [57,58,59]	0.10	--
*Senecio* (Asteraceae)	GC	FID [68,72,77,78,82,84,87,95,99],	--	--
	GC	MS [62,63,67,68,73,74,75,76,77,83,84,87,92,94,99,103,147]	--	--
	GC	NPD [68,72,87],	--	--
	GC	FTIR [71],	--	--
	HPLC	UV [140,151]	0.13 × 10^−3^– 0.31 × 10^−3^	--
	HPLC	ELSD [64]	40	--
	HPLC	MS [86,150]	--	--
	HPLC	MS/MS [66,96,104]	0.5 × 10^−3^	1.0 × 10^−3^
	UHPLC	DAD-MS [61]	--	--
	UHPLC	MS/MS [65]	0.3 × 10^−6^– 11 × 10^−6^	0.8 × 10^−3^–36 × 10^−3^
	None	ELISA [141,163,164,165]	0.02 × 10^−3^– 10,000 × 10^−3^	--
	None	q-NMR (1H/13C) [80,149,166,167]	--	--
*Symphytum* (Boraginaceae)	HPLC	DAD [54]	0.06–0.2	0.10–0.35
	HPLC	ELSD [64]	40	--
	HPLC	MS [116]	--	--
	HPLC	MS/MS [122]	1 × 10–3	5 × 10^−3^
	GC	FID [93,114]	--	--
	GC	MS [114,153]	--	--
	GC	NPD [154]	0.4 × 10–3– 1.0 × 10–3	--
	GC	FTIR [115]	--	--
	TLC	Densiometric [54,168]	22	73
	TLC	Visual [154]	--	--
	None	Photometric [152,169]	1	--
*Tussilago* (Asteraceae)	HPLC	UV [170]	--	--
	HPLC	Q-ToF [117]	0.275 × 10–3	0.916 × 10^−3^
	HPLC	MS/MS [122,171]	<1.0 × 10–3	<5.0 × 10^−3^
	HPLC	MS/MS [157]	0.26–1.32	1.04–5.29
	CE	UV [118]	<0.1 × 10–3	--
	MEKC	UV [121]	2.0 × 10–3– 5.0 × 10–3	--
	GC	MS [120]	--	--
	TLC	VIS [117]	--	--
	Titration	Visually [156]	--	--

**Table 5 toxins-12-00320-t005:** Overview on the studies by Colegate et al. regarding optimal PA extraction for *Eupatorium* [31]. Missing information is characterized by “--”.

	Technique	Sample (g)	Volume (mL)	Time (h)	Temperature (°C)	Solvent	Result (ppm)
1	Tinctures	--	--	--	Room Temperature	Ethanol/Water	Only free bases
2	Maceration	0.2	10	16	Room Temperature	Methanol	Reference Point
3	Infusion	3.3	200	10	Boiling Point.	Water	High contents
4	Decoctions	3.3	200	10	Boiling Point	Water	High contents

**Table 6 toxins-12-00320-t006:** Overview of PA content extracted with two different extraction procedures from *Petasites*. The results correspond to the sum of petasitenine, neopetasitenine, senkirkine, and otosenine [55].

	Technique	Sample (g)	Volume (mL)	Time (h)	Temperature (°C)	Solvent	Result (ratio)
1	Soxhlet	10	--	24	Bp.	Ethanol	1.0
2	Boiling	10	300	1	Bp.	Water	0.5

**Table 7 toxins-12-00320-t007:** Overview on the extraction experiments for Senecio by Hartmann et al. Results correspond to the content of free PA bases determined by GC-MS, and N-oxides by HPLC-UV [72]. Missing information is characterized by “--”.

	Technique	Sample (g)	Volume (mL)	Time (h)	Temperature (°C)	Solvent	Result (%)
1	Maceration	6–10	20	0.5	Room Temperature	Sulfuric Acid 0.1 N	5 *
2	Soxhlet	--	--	48	Boiling Point	Methanol	44 *

* percentage of free bases with reference to the overall PA content.

**Table 8 toxins-12-00320-t008:** Overview of the extraction experiments by Bicchi et al. to optimize the SFE extraction method for *Senecio*. The results correspond to the sum of senecionine and seneciphylline [78]. Missing information is characterized by “--”.

	Technique	Sample (g)	Volume (mL)	Time (h)	Temperature (°C)	Solvent/ Pressure	Result (ppm)
1	Soxhlet	10	--	4	Boiling Point	Methanol ambient	0.74 */2.39 **
2	SFE	0.5	80	4	50	Methanol/ CO_2_ 15MPa	0.68 */2.92 **
3	SFE	0.5	80	4	55	Methanol/ CO_2_ 10MPa	0.65 */2.74 **
4	SFE	0.5	80	4	55	Methanol/CO_2_ 15MPa	0.84 */3.24 **
5	SFE	0.5	80	4	60	Methanol/CO_2_ 15MPa	0.81 */3.16 **

* *Senecio inaequidens* L.; ** *Senecio cordatus* L.

**Table 9 toxins-12-00320-t009:** Overview on the extraction experiments by Kopp et al. using PLE to investigate the influence of different solvents at different temperatures on the extraction yield of PAs for the example of *Senecio* with reference to the BfR-based extraction method. The results correspond to the sum of erucifoline, senecionionine, senecivernine, seneciphylline, retrorsine, and their N-oxides [122].

	Technique	Sample (g)	Volume (mL)	Time (min)	Temperature (°C)	Solvent	Result (ppm)
1	PLE	1	30	30	50/75/100/125	Phosphoric acid 1%	360.6/314.5/331.4/191.2
2	PLE	1	30	30	50/75/100/125	Phosphoric acid 5%	409.7/393.6/396.7/197.6
3	PLE	1	30	30	50/75/100/125	Ammonia 1%	177.9/185.2/218.4/106.6
4	PLE	1	30	30	50/75/100/125	Ammonia 5%	168.9/291.6/212.8/146.3
5	PLE	1	30	30	50/75/100/125	Sulfuric acid 1%	168.4/251.1/119.5/146.5
6	PLE	1	30	30	50/75/100/125	Sulfuric acid 5%	250.7/253.8/85.1/153.8
7	PLE	1	30	30	50/75/100/125	Acetic acid 1%	787.5/863.0/558.2/234.6
8	PLE	1	30	30	50/75/100/125	Acetic acid 5%	831.1/838.0/603.0/195.9
9	PLE	1	30	30	50/75/100/125	Formic acid 1%	798.9/776.9/574.9/255.5
10	PLE	1	30	30	50/75/100/125	Formic acid 5%	880.2/774.4/729.2/208.9
11	BfR based	2	40	30	RT	Formic Acid/Methanol/Water	504.7

**Table 10 toxins-12-00320-t010:** Overview on experiments by Zhang et al. to optimize the extraction of adenofiline by sonication. The results represent the content of adenofiline determined by HPLC-MS/MS [104].

	Technique	Sample (g)	Volume (mL)	Time (min)	Temperature (°C)	Solvent	Result (ppm)
1	Sonication	0.3	10	10	Room Temperature	Methanol/HCl 0.2%	85.2
2	Sonication	0.3	10	20	Room Temperature	Methanol/HCl 1.0%	75.9
3	Sonication	0.3	10	40	Room Temperature	Methanol/HCl 5.0%	85.5
4	Sonication	0.3	25	20	Room Temperature	Methanol/HCl 0.2%	86.6
5	Sonication	0.3	25	40	Room Temperature	Methanol/HCl 1.0%	86.1
6	Sonication	0.3	25	10	Room Temperature	Methanol/HCl 5.0%	74.5
7	Sonication	0.3	40	40	Room Temperature	Methanol/HCl 0.2%	86.8
8	Sonication	0.3	40	10	Room Temperature	Methanol/HCl 1.0%	79.4
9	Sonication	0.3	40	20	Room Temperature	Methanol/HCl 5.0%	81.5

**Table 11 toxins-12-00320-t011:** Overview on the extraction experiments by Mroczek et al. to investigate the influence of extraction techniques using different solvents for different times at different temperatures on the extraction yield of PAs for the example of *Symphytum*. The results correspond to the sum of PAs quantified by UV-Vis [152] (1st, first extraction; 2nd, second extraction).

	Technique	Sample (g)	Volume (mL)	Time (h)	Temperature (°C)	Solvent	Result (ppm)
1	Maceration	1	100	6	Room temperature	Methanol	802
2	Maceration	1	100	12	Room temperature	Methanol	695
3	Maceration	1	100	18	Room temperature	Methanol	854
4	Maceration	1	100	12	50–60	Methanol	1081
5	Reflux	1	100	2 (1st)	Boiling Point	Methanol	1251
6	Reflux	1	100	2 (2nd)	Boiling Point	Methanol	155
7	Percolation	5	500	2	Room temperature	Methanol	1051
8	Percolation	2	200	2	Room temperature	Methanol	640
9	Maceration	1	100	10	Room temperature	1% methanolic solution of tartaric acid	1024
10	Reflux	1	100	1	Boiling Point	1% methanolic solution of tartaric acid	1092
11	Reflux	1	100	2	Boiling Point	1% methanolic solution of tartaric acid	1301
12	Reflux	1	100	4	Boiling Point	1% methanolic solution of tartaric acid	1155
13	Sonication	1	100	0.5	Room temperature	1% methanolic solution of tartaric acid	436
14	Maceration	1	100	18	Room temperature	Ethanol 95%	363
15	Reflux	1	100	4	Boiling Point	Ethanol 95%	1258
16	Maceration	1	100	6	Room temperature	2.5% HCl	498
17	Reflux	1	100	0.5	Boiling Point	2.5% HCl	304
18	Reflux	1	100	2	Boiling Point	2.5% HCl	214
19	Maceration	1	100	1	Room temperature	Chlorofom/MeOH/Ammonia	229
20	Sonication	1	100	0.5	Room temperature	5% Acetic Acid	650
21	Reflux	1	100	2	Boiling Point	1% methanolic solution of ascorbic acid	163

**Table 12 toxins-12-00320-t012:** Overview on the extraction experiments by Feng Liu et al. to investigate the influence of the extraction technique, solvent, extraction time, and temperature on the extraction yield of PAs in *Symphytum*. Results correspond to lycopsamine determined by HPLC- ESI-MS [116].

	Technique	Sample (g)	Volume (mL)	Time (min)	Temperature (°C)	Solvent	Result (ppm)
1	Sonication	1	50	10	Room Temperature	Methanol/Water (50/50)	approx. 7.5
2	Sonication	1	50	10	Room Temperature	Methanol/Water (50/50), pH: 2.5	approx. 7.5
3	Sonication	1	50	10	Room Temperature	Methanol/Chloroform (15/85)	approx. 1.5
4	Sonication	1	50	10	Room Temperature	Methanol	approx. 3.0
5	Sonication	1	50	10	Room Temperature	Ethanol 95%	approx. 1.0
6	Reflux	1	60	60	65	Methanol/Water (50/50)	approx. 30
7	PHWE	1	60	40	60	Methanol/Water (50/50)	approx. 12.5
8	PHWE	1	60	40	80	Methanol/Water (50/50)	approx. 10.0

**Table 13 toxins-12-00320-t013:** Overview on the extraction experiments carried out to investigate the influence of different solvents, at different temperatures on the extraction yield of PAs of *Symphytum* with PLE and the BfR-based extraction method. The result corresponds to the sum of lycopsamine and intermedine quantified by LC/MS/MS [122].

	Technique	Sample (g)	Volume (mL)	Time (min)	Temperature (°C)	Solvent	Result (ppm)
1	PLE	1	30	30	50/75/100/125	Phosphoric acid 1%	389.1/485.5/312.3/615.8
2	PLE	1	30	30	50/75/100/125	Phosphoric acid 5%	564.4/619.5/677.3/586.8
3	PLE	1	30	30	50/75/100/125	Formic acid 1%	321.3/318.4/422.6/462.6
4	PLE	1	30	30	50/75/100/125	Formic acid 5%	474.1/386.1/510.6/472.9
5	PLE	1	30	30	50/75/100/125	Sulfuric acid 1%	481.7/444.2/455.2/486.1
6	PLE	1	30	30	50/75/100/125	Sulfuric acid 5%	726.6/716.4/711.0/502.4
7	BfR based	2	40	30	RT	Formic Acid/Methanol/Water	251.7

**Table 14 toxins-12-00320-t014:** Overview on experiments by Lebada et al. to investigate extraction parameter on the yield of Senkirkine from *Tussilago* quantified by CE-UV [118].

	Technique	Sample (g)	Volume (mL)	Time (min)	Temperature (°C)	Solvent	Result (ppm)
1	Flask	15	1500	30	Room Temperature	Water	2.9
2	Reflux	15	1500	15	Boiling Point	Water	9.3
3	Reflux	10	1000	15	Boiling Point	Water acidified with citric acid	8.0
4	Reflux	10	300	15	Boiling Point	Methanol/Water (50/50) acidified with citric acid	8.0
5	Reflux	10	600	15	Boiling Point	Methanol/Water (50/50) acidified with citric acid	11.2
6	Reflux	10	1000	15	Boiling Point	Methanol/Water (50/50) acidified with citric acid	11.0
7	Reflux	10	1000	120	Boiling Point	Methanol/Water (50/50) acidified with citric acid	9.2
8	Reflux	10	300	15	Boiling Point	Methanol/Water (50/50) alkalized with ammonia	8.9
9	Reflux	10	1000	15	Boiling Point	Methanol/Water (50/50) alkalized with ammonia	8.4
10	Soxhlet	10	500	2880	Boiling Point	Methanol	2.5
11	Reflux	10	1000	15	Boiling Point	Methanol alkalized with ammonia	5.0

**Table 15 toxins-12-00320-t015:** Overview on the extraction experiments by Jiang et al. to investigate the influence of different extraction techniques on the yield of senecionine and senkirkine extracted from *Tussilago*. The result correspond to the sum of senecionine and senkirkine determined by LC/MS/MS [157].

	Technique	Sample (g)	Volume (mL)	Time (min)	Temperature (°C)	Solvent	Result (ppm)
1	MAE	1	40	15	Bp.	Water/Methanol acidified with HCl	104.4
2	Reflux	1	60	60	Bp.	1 N HCl	109.6
3	PHWE	0.25	50	50	60	Water	88.2
4	Reflux	1	60	60	Bp.	1 N HCl	87.9

**Table 16 toxins-12-00320-t016:** Overview on the extraction experiments investigating the influence of different solvents, at different temperatures on the extraction yield of PAs for the example of *Tussilago* with reference to the BfR-based extraction method. The result corresponds to the sum of senkirkine, senecionine, and senecivernine quantified by LC/MS/MS [122].

	Technique	Sample (g)	Volume (mL)	Time (min)	Temperature (°C)	Solvent	Result (ppm)
1	PLE	1	30	30	50/75/100/125	Phosphoric acid 1%	62.3/63.6/62.0/62.8
2	PLE	1	30	30	50/75/−0/125	Phosphoric acid 5%	58.3/64.8/65.0/62.4
3	PLE	1	30	30	50/75/100/125	Ammonia 1%	21.0/21.5/22.2/20.9
4	PLE	1	30	30	50/75/100/125	Ammonia 5%	21.2/20.8/20.2/20.6
5	PLE	1	30	30	50/75/100/125	Sulfuric acid 1%	60.6/62.0/65.6/63.4
6	PLE	1	30	30	50/75/100/125	Sulfuric acid 5%	60.9/61.2/60.0/62.8
7	PLE	1	30	30	50/75/100/125	Acetic acid 1%	59.7/62.4/62.4/62.8
8	PLE	1	30	30	50/75/100/125	Acetic acid 5%	63.7/60.7/62.6/63.7
9	PLE	1	30	30	50/75/100/125	Formic acid 1%	62.8/62.0/63.3/64.3
10	PLE	1	30	30	50/75/100/125	Formic acid 5%	59.6/62.6/62.2/63.6
11	BfR based	2	40	30	RT	Formic Acid/Methanol/Water	41.0

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
