# Peer review of "Extracting and Analyzing Pyrrolizidine Alkaloids in Medicinal Plants: A Review"

_toxins, 2020, doi:10.3390/toxins12050320_

Round 1
Reviewer 1 Report
This review of pyrrolizidine alkaloids (PAs) is well written and complete. Its extensive use of tables with allows for easy reference of extraction and analytical techniques.
I have only two comments. It might be useful for the reader if the authors included images of the plants they discuss along with relevant chemical structures of the PA mentioned in the review.
Author Response
See attached document.

Reviewer 2 Report
The manuscript titled “Extracting and Analyzing Pyrrolizidine Alkaloids in 2 Medicinal Plants: A Review” is a systematic literature research, carried out with the aim of improving the knowledge on the relationship between extraction methods and toxicity of pyrrolizidine alkaloids. The paper is well presented and properly discussed.
In my opinion, it would be useful to add in the introductory section a mention of the anticholinesterase activity that many of these alkaloids possess (and which seems related to the defense mechanisms against insects). In particular, I would add some references to the bibliography, given below.
- Schmeller, T.; El-Shazly, A.; Wink, M. Allelochemical activities of Pyrrolizidine Alkaloids: interactions with neuroreceptors and Acetylcholine related enzymes. Journal of Chemical Ecology. 1997, 23, 2, 399-416.
- Rietjens, I.M.C.M.; Martena, M.J.; Boersma, M.G.; Spiegelenberg, W.; Alink, G.M. Molecular mechanisms of toxicity of important foodborne phytotoxins. Mol. Nutr. Food Res. 2005, 49, 131–158. DOI 10.1002/mnfr.200400078
- Benamar, H.; Tomassini, L.; Venditti, A.; Marouf, A.; Bennaceur, M.; Serafini, M.; Nicoletti M. Acetylcholinesterase inhibitory activity of pyrrolizidine alkaloids from Echium confusum Coincy. Natural Product Research 2017, 31, 11, 1277–1285. DOI: 10.1080/14786419.2016.1242000
- Moreira, R.; Pereira, D.M.; Valentão P.; Andrade P.B. Pyrrolizidine Alkaloids: Chemistry, Pharmacology,
Toxicology and Food Safety. Int. J. Mol. Sci. 2018, 19, 1668. DOI 10.3390/ijms19061668
Furthermore, it should be remembered, speaking of the genus Cynoglossum, that it has been heavily modified from a systematic point of view in recent times, given the strong affinity of some of its species with neighboring genera (such as Pardoglossum and Solenanthus).
- Selvi F.; Coppi, A.; Cecchi,L. High epizoochorous specialization and low DNA sequence divergence in Mediterranean Cynoglossum (Boraginaceae): Evidence from fruit traits and ITS region. Taxon 2011, 60, 4, 969–985. DOI: 10.1002/tax.604003
- Hartmut H. Hilger, H.H.; Greuter, W.; Stier, V. Taxa and names in Cynoglossum sensu lato (Boraginaceae, Cynoglosseae): an annotated, synonymic inventory, with links to the protologues and mention of original material. Biodivers Data J. 2015, ,3, e4831. DOI 10.3897/BDJ.3.e4831
Author Response
See attached document

Reviewer 3 Report
The paper describes the analysis of pyrrolizidine alkaloids from specific herbs. It looks like a thorough review of the literature, which by itself is not an easy task regarding the many different plant species and techniques applied. This also makes the paper a bit difficult to read and there are some possibilities to improve this. It would e.g. be helpful to add a short description of different extraction and analytical techniques before describing the different plant species. Some of the terms are known by specialists but not in general. Also explain here the abbreviations used later (like PLE, PHWE etc.). Maybe in this part a reference can be made to Tables 2 (extraction) and 3 (analysis), since this is not consistently done later on. Somewhere it may also be discussed if extraction methods have to be tested for each type of plant or whether the results from one can be extrapolated to others.
The authors do mention that the issue of PAs is not only with respect to specific herbs but also concerns contamination of other foods with PA containing herbs. In terms of levels this requires much more sensitive methods and maybe this could be stressed somewhere, also in relation to typical levels in herbs vs contaminated food items like honey and tea.
Line 10: are acute effects expected at such low exposures or ws it based on genotoxic and carcinogenic properties?
Line 12 and line 46: exhaustive sounds a bit strange: effective or efficient?
Line 14: phytopharmaceutical ...: something seems missing; what is meant by timely?
Line 17: 1110 ppm (preference for mg/kg. And in this case a % would be more useful.
Line 19: the paper shows that some methods work better than others. Are still other methods needed or is it more a matter of standardization of extraction methods? And it is not really discussed what still should be improved in the analysis. Would that include more standards or use of methods that look at the necine bases, or e.g. effect-based assays in combination with chemical analysis? This is also related to how this should lead to better safety, in particular for herbs that seem the topic of this review, more than contaminated food items like tea and honey.
Line 27: its efficiency; its seems to refer to pyrrolizidine alkaloid content, but if so, what does efficiency mean?
Line 33: genotoxic by itself is less the issue than the liver tumours observed in animals, which combined with genotoxicity means that the exposure should be as low as possible. EFSA did e.g. not derive an HBGV for PAs but used a Margin of Exposure approach using carcinogenicity data from a study with riddelliine.
Line 38: exposure?
Line 39: Higher PA contents (10 µg/day) sounds strange since the latter reflects exposure and not a content. Please rephrase like: contents leading to high exposure.
Line 45: 0.42 µg/day seems higher (not lower) than the 0.1 µg/day mentioned in line 39 for products with longer exposure. Was this for contaminated products (like tea) or only for herbs?
Line 57: the period sounds very short and later on this is mentioned as not restricted to this period, which is supported by the reference list. What is meant? Were more papers retrieved based on snowballing, i.e. references mentioned in other papers?
Line 73: it seems a bit more than 14 plants: species or genera?
Line 85: and throughout the paper: et al., so not et.al.
Line 92: does this include borago officinalis which is also used in food?
Line 99: their rather than her (plural)
Line 103: visualizing them ..
Line 107: headings in table 1 seem to be missing; better not to use parallel columns with plant species.
Line 108: the family name for Tussilago (Asteraceae?) seems missing
Table 2: also here, the question is if the table is easier to read when putting every family on a separate row (like table 3). Not sure if the table will be much longer. Some edits: dichloromethane, petroleum. What is meant by Et2O?
Table 3: ToF-MS, several places? Make second column a bit broader to reduce the length.
Line 139: can the statement about sensitivity be moved to the suggested more general part since it applies for most plants families mentioned.
Line 151: what is the difference between alcohol tinctures and tinctures in this sentence, since this is not clear from table 4.
Line 173: Heliotropium; applied rather than investigated
Line 178: have a positive...
Line 186: GC-MS
Line 203: PAs?
Line 213: why however; actually that word is quite often used and not always needed; see also line 221 and 282
Table 5 (also other tables): why is there no volume for Soxhlet: not mentioned in the paper? If so, please indicate that.
Line 228: In general or in most cases, not both
Line 244: for rather than due
Line 250: mutagenic seems strange in this list of positive effects
Table 6: can also the measured levels be shown, so not only %?
Line 268: not sure if pressure is more important than temperature based on Table 7 (compare e.g 4 vs 2: 1.2/1.1 fold increase as compared to 1.3 and 1.2 for 4 vs 3.
Line 281: of up to over?
Table 8: maybe better to use no digits for the levels
Line 289: correspond to
Line 295: concluded rather than figured out.
Table 9: differences seem rather small and based on the large number of variables, it seems difficult to draw firm conclusions. Maybe this study can briefly be described but without a table?
Line 308: individual: what is meant?
Line 308: HPLC-based analyses?
Line 323: one is too many
Table 10: 2.5 rather than 2,5
Line 361: what is meant by the BfR method? In several places. Was this always included as a reference by the authors or was the same material tested by BfR?
Line 371: was shown
Line 377: has high
Line 382: reached
Line 396: Table 2, where maceration
Line 403: but also 1 g/60 ml (no 5)?
Line 407: but only tested for Water, correct?
Line 426: what is meant?
Line 447: for rather than of; N-oxides
Line 476: again, a percentage seems more informative than just the difference
Line 479: best rather than increased?
Line 481: what about alkaline?
Line 503: FID or UV, respectively (suggestion correct?)
Line 504: one is too many
Line 512: unclear what is meant
Line 523: PAs
Line 513: in 2016 again decreased: in the introduction 2011 is mentioned. Also there is a RP from JECFA and EFSA based on Riddelliine; can that be mentioned as well?
Author Response
See attached document

Round 2
Reviewer 3 Report
The revised document is much improved. Some remaining issues:
- Line 17: plant families? Or plant genera (as in line 43. Also line 62, 80 etc.: plant genera.
- Line 38: is this neurotoxicity really correct (what about humans using these supplements)? If so, should it be: due to their anticholinesterase activity or inhibition of cholinesterase activity?
- Line 39: good to see that carcinogenicity is included but then it is absent from the next line. Genotoxicity and cytotoxicity are more modes of action and not the final adverse outcomes. And here neurotoxicity is lacking, if that would really be an effects of these compounds.
- Line 55: the suggestion is made that the NTP studies on lasiocarpine (1978) and riddelliine (2003) were new but they were already available in 2011. Maybe it is meant that the risk assessments were new. It is always a bit tricky to explain the use of an RP for genotoxic carcinogens, as ideally the exposure is zero (no threshold). In practice an MOE of 10,000 is used. The new RP used by EFSA was the one based on riddelliine (NTP, 2003) and was a BMDL10 of 237 µg/kg bw/d. This means that an exposure lower than 0.0237 µg/kg/d results in a Margin of Exposure larger than 10,000 and would thus be of low health concern. It does not necessarily mean that lower exposures are acceptable. So the RP is actually 30.000 fold higher than the 0.42 µg/day. Maybe say: New risk assessments by EFSA and BfR were based on the NTP study on riddelliine rather than the one on lasiocarpine and derived a 3-fold higher reference point of 237 µg/kg bw/day. This effectively means that an exposure lower than 0.0237 µg/kg bw/day is considered a low health concern (Margin of Exposure larger than 10,000). This intake is about three-fold higher than the one used before. In the next line, the start: despite the increased limits, seems not ideal. It would be strange if this three-fold increase would suddenly change everything. Furthermore, that really depends on the MLs which are not always based on the required levels but rather on the principal strict but feasible (kind of ALARA). Suggestion to start: Especially for the required detection of the low levels of PAs in contaminated products, .... Still it should be explained somewhere that this review focusses on the analysis of PA-containing plants and not contaminated products, since the required LoQs are much higher.
- Line 70: what is TK?
- Line 90 mentioned/discussed; line 99 augmented? Is this the start of a new sentence since what follows seem separation techniques again and not detection methods.
- The order of the tables seems not correct: should it be first the table 2 with identified PAs based on chapter 3, followed by the tables on extraction techniques (now 1), solvents and analytical detection methods (chapter 4). And the figures are already mention in chapter 3 but some of them occur at a very late stage.
- Line 594: how can a content be expressed in µg/day? Please rephrase. Was the content actually regulated, meaning, were limits set for the levels in the plants? This would be interesting to mention as it shows if LoQs are low enough. Line 595, kg bw? And should it be: BfR recommended a maximum intake of ...., which meant a reduction by about 50%? (not sure if the rest of the sentence is needed, also because the study used at that time was from 1978, so not really new). And for the next sentence, see above: new risk assessments, not new studies. And without explaining that the exposure should be at least 10,0000 -fold lower, this value cannot easily be compared. And the reference point is not a limit: EFSA doesn’t derive HBGVs for genotoxic carcinogens; not sure about BfR. Not sure if this section on risk assessment should be repeated here or that there can be a general statement that sensitive methods are needed, especially for contaminations of regular food products.
- Could the et.al. still contain 1 dot too many since et is not an abbreviation? (throughout the paper).
Author Response
See attached document.
